# Embryo Culture, In Vitro Propagation, and Molecular Identification for Advanced Olive Breeding Programs

Vito Montilon [1], Leonardo Susca [1], Oriana Potere [1], Vincenzo Roseti [1], Antonia Campanale [2], Antonia Saponari [3], Cinzia Montemurro [1,2,4], Valentina Fanelli [1], Pasquale Venerito [3] and Giovanna Bottalico [1,*]

1 Department of Soil, Plant and Food Sciences (DiSSPA), University of Bari Aldo Moro, 70126 Bari, Italy; vito.montilon@uniba.it (V.M.); leonardo.susca@uniba.it (L.S.); oriana.potere@uniba.it (O.P.); vr.plants@gmail.com (V.R.); cinzia.montemurro@uniba.it (C.M.); valentina.fanelli@uniba.it (V.F.)
2 Institute for Sustainable Plant Protection-IPSP—CNR, 70126 Bari, Italy; antonia.campanale@ipsp.cnr.it
3 Centre for Research, Experimentation and Education in Agriculture (CRSFA) "Basile Caramia", 70010 Locorotondo, Italy; antonellasaponari@crsfa.it (A.S.); pasven04@libero.it (P.V.)
4 Department of Agricultural and Environmental Science (DiSAAT), Sinagri Spinoff of University of Bari Aldo Moro, 70126 Bari, Italy
* Correspondence: giovanna.bottalico@uniba.it; Tel.: +39-080-544-3418

**Abstract:** The high biodiversity of the olive tree is an important opportunity to develop sustainable plans to control *Xylella fastidiosa* (*Xf*) through breeding programs. Olive tree breeding activities have been limited due to various features of this species including the long time required for seed germination caused by the inhibition effect of the woody endocarp, the seed integument, and the endosperm. Starting from $F_1$ seeds by cross-breeding, the embryo culture was compared with traditional seed germination, evaluating the effectiveness of in vitro multiplication of the plantlets for large-scale production. The isolated embryos were established on a new medium based on Rugini '84 macroelements, Murashige & Skoog '62 microelements, with Nitsch J. P. & Nitsch C. '69 vitamine and subcultured on Leva MSM modified. The results obtained confirmed that in vitro culture of olive embryos is a valid tool for increasing the percentage and speed of germination, helping to reduce the time of the olive breeding programs, offering the possibility to effectively propagate plantlets for further experiments.

**Keywords:** seed germination; embryo culture; clonal propagation; olive tree; SSR markers; intervarietal crossbreeding

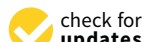



## 1. Introduction

Olive is a very important and widespread species in the Mediterranean area, it is cultivated mainly for fruit harvesting, used for oil production or table consumption exclusively, or for both. Due to the prevalent outcrossing nature of the species [1], olive has a high level of genetic variability distributed among the numerous cultivars and within the different accessions [2]. This variability represents an important resource that can be used in breeding programs. Olive breeding is essentially based on clonal selection and cross-breeding. Clonal selection represented the main method for developing new varieties, and it is based on the selection of clones showing the desired traits, allowing preservation of the sanitary quality through the selection of plants free from viral agents [3,4] and the propagation material within a certification system [5]. The main traits of interest are adaptation to modern cultivation systems, the production of a higher quality olive oil, and the resistance to pathogens. In the last five years, olive breeding programs have acquired great importance due to the opportunity to increase the quantity and quality of production, reduce vegetative vigor for adaptation to mechanical operations, decrease alternate bearing, increase the self-fertility rate, and improve rooting ability. In addition, the resistance to unfavorable environmental conditions such as cold, salinity, drought, or the main pests,

including *Verticillium dahliae*, *Mycocentrospora cladosporioides*, *Bactrocera oleae*, *Prays oleae*, and *Pseudomonas syringae* pv. *savastanoi* are also highly desirable traits [6,7]. The recent outbreak of the Olive Quick Decline Syndrome (OQDS), caused by *Xylella fastidiosa* subsp. *pauca* (*Xf*), is dramatically altering ecosystem services in the peninsula of Salento, in the Apulia region in southern Italy. Replanting with resistant genotypes appears to be the most feasible and promising strategy to control the bacterium. So far, the cultivars 'Leccino' and 'FS-17' have shown resistance to *Xf* [8,9]. Therefore, new and quick breeding activities to expand olive biodiversity merging *Xf* resistance to other economically important traits, such as agronomic performance and oil quality, are needed [10]. Multiplication by seed is essential in traditional genetic improvement enabling variability and production of new cultivars. In olive, the seed germination occurs gradually, over a long time, and at low rates [11]. These factors represent some of the conditions that have limited the breeding activities of this species, together with several other unfavorable aspects such as the self-incompatibility of many cultivars, the low fruit set rates, and the long duration of the juvenile stage [12,13]. In previous studies, the inhibitory effect of woody endocarp, seed integument, and endosperm on the olive seed germination was described [14,15]. By removing the woody endocarp, the germination capacity remains still low and scalar [11,16]; moreover, treatments with exogenous phytohormones were not very effective or only partially satisfactory to break dormancy [17]. With the aim to achieve higher and uniform levels of germination rate and reduce timing by breaking mechanical dormancy, new protocols for in vitro embryo culture were developed [18,19]. Embryo culture refers to one of the applications of plant tissue culture widely used in genetic improvement to save the hybrid products of fertilization that might otherwise degenerate (properly called embryo rescue) or to overcome dormancy in ripened seeds [20,21]. This technique consists in the cultivation of zygotic embryos excised from ovules and seeds under aseptic conditions on a sterile nutrient medium [22]. The embryogenic cultures could also be maintained in vitro and clonally propagated or used to perform tests for the selection of interesting genotypes. Indeed, in vitro cultures have been exploited in screening works for salt and drought tolerance and biotic stress resistance in several species [23]. This work reports an efficient and rapid protocol for in vitro culture of isolated embryos to produce and propagate the olive plantlets that was developed in the frame of a breeding program to obtain olive genotypes having superior resistance to *Xf*. We also tested the efficacy of a new embryo culture medium compared to another just validated. Furthermore, in vitro experiments for multiplication of selected plant material were performed.

## 2. Materials and Methods

### 2.1. Plant Material

Controlled crossbreeding activities were carried out in the years 2017 and 2019 on plants grown in an ex situ conservation field of the olive germplasm collection at the Centre for Research, Experimentation, and Education in Agriculture "Basile Caramia" in Apulia, South Italy. The parents were chosen including some commercial and widespread varieties in Italy, particularly Leccino, FS 17, Cellina di Nardò, Cima di Melfi, Coratina, and Carolea.

### 2.2. DNA Extraction and Microsatellite Analysis

The identity of each parent was confirmed through molecular fingerprinting using highly polymorphic simple sequence repeat (SSR) markers that were selected on the basis of studies previously performed [24,25]. DNA was extracted from fresh young leaves following the protocol of Spadoni et al. [26]. Extracted DNA was checked for quantity and quality through a Nano-Drop™2000C Spectrophotometer (Thermo Scientific, Waltham, MA, USA) and 1% agarose gel electrophoresis. Samples were genotyped using a set of 10 SSR markers (DCA03, DCA05, DCA09, DCA13, DCA17, DCA18, GAPU71b, GAPU101, EMOL, and EMO90) previously reported to be effective in differentiating olive genetic resources [24]. A PCR reaction was performed following Spadoni et al. [26]. Each amplification product (2 μL) was mixed with 14 μL of formamide and 0.5 μL of the GeneScan

600 LIZ size standard (Life Technologies, Carlsbad, CA, USA) and then analyzed using the ABI PRISM 3100 Avant Genetic Analyzer automatic sequencer (Applied Biosystems, Foster City, CA, USA). The allele sizes were assigned through the GeneMapper Software version 3.7 (Life Technologies, Carlsbad, CA, USA). The molecular profile of each parent was compared with those available from previous studies [25,27,28].

### 2.3. Controlled Crossings

Controlled crossings were conducted following the procedure previously described [29]. The fruits were harvested in late summer (September), at the veraison, when the color of the drupes turned violet, reported as associated with a high level of viable seedlings [30–32]. Subsequently, the flesh (mesocarp) was removed mechanically with a blender to recover the olive stones (endocarp) that were washed and dried in the open air for one day. Then, the stones were treated with broad-spectrum fungicide and stored in bags at 4 °C for three months to break the physiological dormancy. After cold storage, the stones were used either for direct sowing or embryo culture.

### 2.4. Traditional Sowing

Direct sowing was performed only for the seeds obtained from the cross-breeding carried out in 2017. In detail, the seeds were taken out from the cold chamber and immersed in tap water for a week renewing the water daily, then treated in a broad-spectrum fungicide solution for one hour before sowing. The sowing was performed in seedbeds containing a mixture of peat, soil, and sand inside a thermo-conditioned greenhouse where temperature averaged approximately 20/30 °C, with relative humidity at 40/70%, and natural light. A total of 4160 olive stones were directly sown resulting from 5 cross combinations. Germination was determined after five months after sowing, when over 95% of the seedlings emerged for all combinations and the number of plants obtained was recorded to calculate the germination percentage.

### 2.5. In Vitro Embryo Culture

For embryo culture two alternative substrates were compared, named 'medium 1' and 'medium 2', followed by 2–3 subcultures to obtain further plants. After the cold treatment, the seeds were extracted by breaking the woody endocarp in a bench vise [11], then stored in a refrigerated cell at 4 °C until processing within a short period of time. Subsequently, the seeds were surface sterilized by immersion in a 70% ($v/v$) ethanol absolute solution for 3 min, followed by soaking for 20 min in a 20% ($v/v$) sodium hypochlorite solution. Finally, explants were rinsed in sterile water three times. The sterilized seeds were stored under aseptic conditions for 48 h in Petri dishes containing filter paper soaked in sterile distilled water to allow the imbibition of seeds and easier extraction of embryos. At the time of in vitro establishment, a second sterilization of the seeds was performed by dipping the seeds in a 20% sodium hypochlorite solution for 3 min, followed by 3 washes in sterile water. The embryos were extracted without injury with sterile forceps and scalpel under the stereomicroscope and transferred individually or in small groups of 2–3 embryos, into test tubes or Petri dishes containing each 10 mL, either of media 1 or 2. The tubes were then sealed with parafilm and incubated in a growth chamber at a constant temperature of 27 °C and 16 h photoperiod, exposed to white led light at 5500 °K.

From the crosses performed in the 2017 season, 216 embryos were obtained in total, of which 126 were grown on medium 1 and 90 on medium 2. All seeds from the 2019 crossbreeds were cultured on olive embryo medium 1 exclusively; in detail, 143 and 138 embryos for combinations 1 and 2, were established in vitro.

The germinability was determined after three weeks by calculating the percentage of embryos with the radicle reached a length of at least 0.3 cm and the open cotyledons as reported by Germanà et al., 2014 [33], on the number of olive stones. The comparison between the two media was performed by determining the germinability on the number of

in vitro successfully stabilized embryos obtained from the difference between the initial number of embryos cultured and the number of embryos lost due to contamination.

Germinated embryos were transferred on an in vitro olive multiplication medium for further growth. We reached an adequate development of the foliar and root system that occurred after about eight weeks from the in vitro establishment, the plantlets were extracted and divided into two parts: one of them, (1 cm of apical part) in vitro, was subjected to serial subcultures of 30–40 days each, the other one (the rooted part) was transplanted in vivo.

This part was washed under tap water to remove the adhering medium and transplanted to pots containing a mixture of sterilized peat, soil, and sand for the acclimatization step in an air-conditioned room set to a temperature of 27 °C and a 16 h photoperiod exposed to 3000 °K led light (T.S.A. Technology s.r.l., Serravalle, Republic of San Marino). The survival rate of acclimatized plantlets was also recorded.

The microshoots obtained from the first subculture of apices were divided into uninodal cuttings and individually propagated on 40 mL of the multiplication medium into glass jars or plastic containers. In the subsequent subculture cycles, the induced microshoots and axillary buds were split and subcultured again. The proliferation of shoots from the basal callus was also occasionally obtained. After each cycle of subculture, the axillary and adventitious shoots were counted; moreover, the length of the shoots and the number of nodes were determined (data not shown). These data were averaged across all seven cross combinations to determine the multiplication rate as the mean number of potentially inducible shoots starting from a single explant after each subculture cycle [34].

### 2.6. Media Composition

In this work, two olive embryos media were compared. Medium 1 was tested as a new formula without hormones and with 1/2 strength of the macroelements OM [34] and microelements MS [35], vitamin NN [36], modified with 36 g $L^{-1}$ of D-mannitol, and 7 g $L^{-1}$ of plant agar (Duchefa Biochemie, Haarlem, The Netherlands).

Medium 2, whose mineral composition consisted of macro and microelements MS [35], was added to 1 mg $L^{-1}$ of zeatin riboside, 36 g $L^{-1}$ of D-mannitol, and 7 g $L^{-1}$ of plant agar (Duchefa). In the multiplication step, the medium MSM [37] modified with 1 mg $L^{-1}$ zeatin riboside and 7 g $L^{-1}$ of plant agar (Duchefa Biochemie, Haarlem, The Netherlands) was used.

All media were sterilized at 120 °C for 20 min, and the pH was adjusted to 5.8 before autoclaving.

### 2.7. Statistical Analysis

The statistical analysis, to compare the media, was carried out as a completely randomized design. Germinability data were analyzed by means of the Pearson's chi-square test performed by using the open-source software RStudio 4.1.0 (18 May 2021) with 5% significance.

## 3. Results

### 3.1. SSR Markers Analysis

A perfect match between the analyzed samples and the corresponding variety was found, confirming the identity of each parental cultivar used in the controlled crossings (Table 1).

### 3.2. Traditional Sowing

The germination of the olive stones, sown directly in the seedbed, began three months after the sowing and continued gradually, with over 95% of all seedlings emerging after five months as reported in Table 2.

**Table 1.** Allelic profile obtained for each parental olive cultivar using 10 SSR markers. Allele size is reported in base pairs.

| Parental Cultivar | DCA03 | | DCA05 | | DCA09 | | DCA13 | | DCA17 | | DCA18 | | GAPU71b | | GAPU101 | | EMO90 | | EMOL | |
|---|---|---|---|---|---|---|---|---|---|---|---|---|---|---|---|---|---|---|---|---|
| Cellina di Nardò | 232 | 239 | 206 | 214 | 186 | 204 | 124 | 124 | 179 | 179 | 171 | 171 | 124 | 127 | 192 | 198 | 188 | 190 | 192 | 192 |
| Cima di Melfi | 243 | 253 | 198 | 206 | 186 | 204 | 156 | 156 | 143 | 143 | 175 | 177 | 124 | 144 | 182 | 182 | 188 | 188 | 198 | 198 |
| Carolea | 232 | 253 | 194 | 206 | 162 | 198 | 122 | 140 | 115 | 179 | 179 | 181 | 121 | 130 | 192 | 218 | 188 | 198 | 198 | 198 |
| Coratina | 239 | 243 | 198 | 206 | 182 | 194 | 120 | 156 | 115 | 115 | 177 | 181 | 124 | 144 | 198 | 218 | 188 | 194 | 198 | 198 |
| FS 17 | 232 | 245 | 206 | 208 | 206 | 206 | 120 | 120 | 115 | 143 | 173 | 177 | 124 | 144 | 198 | 198 | 190 | 194 | 192 | 198 |
| Leccino | 243 | 253 | 198 | 206 | 162 | 206 | 120 | 120 | 107 | 117 | 177 | 177 | 124 | 144 | 198 | 200 | 188 | 194 | 198 | 198 |

**Table 2.** Germination rates of the olive stones directly sown for the different cross combinations performed.

| Crossbreed Combination | No. of Stones Directly Sown | No. of Seedlings Emerged after 5 Months | Germinability (%) |
|---|---|---|---|
| Combination 1—Cima di Melfi × FS 17 | 818 | 131 | 16.01 |
| Combination 2—Leccino × Cima di Melfi | 364 | 53 | 14.56 |
| Combination 3—Leccino free pollinated | 571 | 89 | 15.59 |
| Combination 4—Leccino × Cellina di Nardò | 686 | 104 | 15.16 |
| Combination 5—Leccino × FS 17 | 1721 | 416 | 24.17 |

The germinability ranged from 14.56%, observed in combination 2, to 24.17% in combination 5, with an average of 17.10%.

### 3.3. In Vitro Embryo Culture

The culture of the isolated embryos allowed the achievement of a significantly higher, faster, and homogeneous germination rate than the direct sowing of olive stones on both tested media and for all the cross combinations. The germination of the embryos started after about five days with divergence and greening of the cotyledons (Figure 1a), considered the first visible signs of germination [16]. After ten days most of the embryos reached a radicle development of over 0.3 mm and had the cotyledons opened (Figure 1b). The percentage of germination was determined after three weeks when almost the maximum value of germination rate was reached (Figure 1c). In 2017, the percentage of embryos germinated on media 1 and 2, calculated on the total olive stones, was 57.22% and 51.67%, respectively. In the reference period, a relatively high frequency of empty olive stones was obtained. Indeed, 54 and 30 aborted seeds were registered on 180 and 120 olive stones for media 1 and 2, respectively, corresponding to frequencies of 30% and 25%. The germinability determined on 126 and 90 cultured embryos was of 81.75% for medium 1 and 68.89% for medium 2 with an average value of 75.32%.

The number of successfully stabilized embryos for media 1 and media 2 was determined by the difference between 126 and 90 embryos cultured minus 14 and 20 embryos lost due to contamination, respectively, corresponding to 112 and 70 embryos. The germinated embryos on 112 and 70 in vitro successfully stabilized embryos, resulting in germination rates of 91.96% and 88.57% for media 1 and 2, respectively. Statistical analysis indicated no significant difference between the germination rates on the two media tested ($\chi^2 = 0.25345$, 1 df, *p*-value = 0.6147). The germination percentages detected on the two types of media tested related to the isolated embryos of 2017 are reported in Table 3. Germination percentages of the in vitro isolated embryos compared to the germinability of traditional sowing relative to the 2017 season are represented in Figure 2.

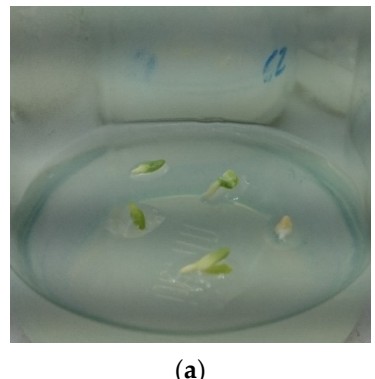  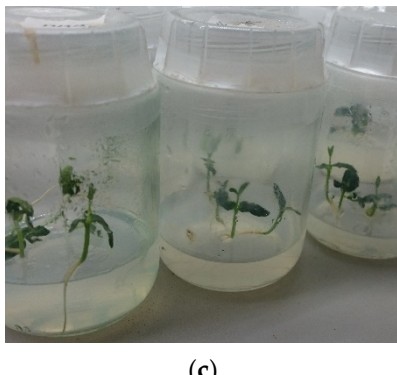

(**a**)        (**b**)        (**c**)

**Figure 1.** (**a**) Isolated olive embryos in vitro established after 5 days; (**b**) Germination of embryos after 10 days from establishment; (**c**) Plantlets developed from embryos at 21 days of cultivation.

**Table 3.** Germination rates detected on the two types of media tested related to isolated olive embryos of the cross combination 5 (Leccino × FS 17) of the 2017 season.

| Medium Type | No. of Stones | No. of Embryos Cultured | No. of Embryos Lost Due to Contamination | No. of Ungerminated Embryos | No. of Germinated Embryos | [1] Germinability (%) on the Total Olive stones |
|---|---|---|---|---|---|---|
| Medium 1 | 180 | 126 | 14 | 9 | 103 | 57.22 |
| Medium 2 | 120 | 90 | 20 | 8 | 62 | 51.67 |

Statistical analysis of the germinability of isolated embryos between the two media tested

| | no. successfully stabilized embryos | no. of germinated embryos | [1,2] germinability (%) on the successfully stabilized embryos |
|---|---|---|---|
| Medium 1 | 112 | 103 | 91.96% |
| Medium 2 | 70 | 62 | 88.57% |
| $\chi^2$ value | 0.25345 | | |
| *p*-value | 0.6147 | | |

[1] Data on germination were detected after 3 weeks of culture. [2] Olive embryo germinability between the two media tested was not significantly different (*p* = 0.05) using the chi-square test.

In 2019, the number of empty olive stones detected for the crossbreeds 6 and 7 was 57 and 62, respectively. The germinated embryos of the combinations 6 and 7 were 109 and 105 on a total for each of 200 stones, with the corresponding germinability of 54.5% and 52.5%, respectively as reported in Table 4. The germination rates referred to 143 and 138 cultured embryos of crosses 6 and 7, which were 76.22% and 76.09% with an average value of 76.15%.

**Table 4.** In vitro germinability, after 3 weeks of culture, of isolated olive embryos related to the cross combinations performed in the 2019 season.

| Crossbreed Combination | No. of Stones | No. of Embryos Cultured | No. of Ungerminated Embryos | No. of Germinated Embryos | No. of Embryos Lost Due to Contamination | Germinability (%) on the Total Olive Stones |
|---|---|---|---|---|---|---|
| Combination 6—FS 17 × Coratina | 200 | 143 | 13 | 109 | 21 | 54.5 |
| Combination 7—FS 17 × Carolea | 200 | 138 | 11 | 105 | 22 | 52.5 |

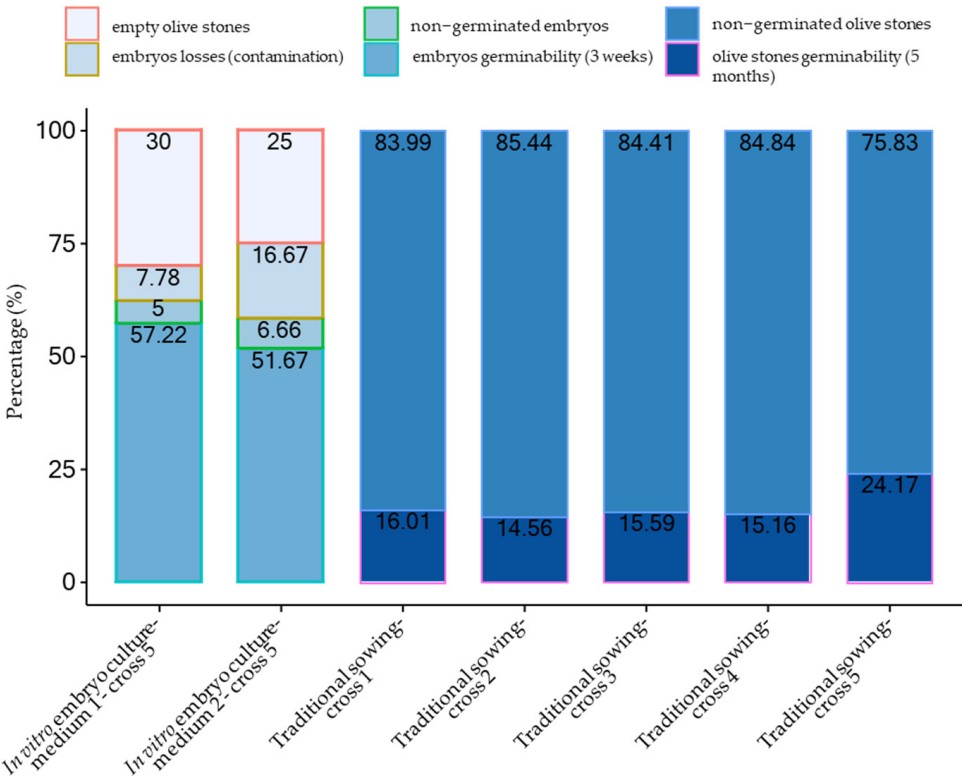

**Figure 2.** Germination percentages of in vitro isolated olive embryos on the two media tested compared to the germinability of directly sown olive stones from the 5 different cross combinations of the 2017 season.

All embryos exhibited normal growth and development, without growth alterations such as hyperhydricity, or they have occurred rarely, during the in vitro culture of the apical part (Figure 3a) as after transplantation and acclimatization of the bottom part of the same plants in the greenhouse (Figure 3b). The survival rate of the plantlets in the acclimatization phase was higher than 90% for all combinations. Moreover, in vitro cultured plantlets showed a good rooting attitude with frequent emission of roots when grown on the multiplication substrate without hormones.

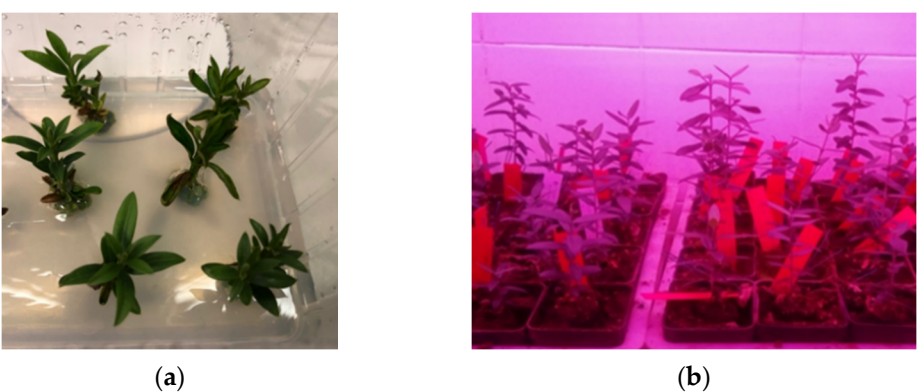

(**a**)                                    (**b**)

**Figure 3.** (**a**) In vitro olive plantlets in the multiplication step; (**b**) plantlets developed from isolated embryos during the acclimatization phase in the growth chamber.

Considering all combinations, the shoot cultures were multiplied by about 2.28 fold after each subculture cycle lasting about 30–40 days. The theoretical number of explants that can be produced from a single plantlet after *n* subculture cycles may be estimated raising the multiplication rate to the *n* number of subcultures. Based on the results obtained and

assuming nine subcultures per year, it is estimated that it is possible to produce indicatively $2^9$ = 500 explants in one year (where ca 2, the multiplication rate obtained, was raised to nine subcultures).

## 4. Discussion

The high level of genetic variability present in olive trees can be exploited in breeding programs aiming to improve the adaptation to different cultural conditions or for resistance to pathogenic organisms. Currently, OQDS is regarded as one of the most destructive plant disease in Apulia, South Italy. It is caused by the xylem-limited bacterium *Xylella fastidiosa* subsp. *pauca* (*Xf*). So far, the olive tree cultivars Leccino and FS17 represent natural sources of resistance to the disease [8,9], which can be exploited directly in the field or in breeding programs with the aim to obtain genotypes useful for the future evaluation of *Xf*-resistant olives. The overall objective of the present work was to propose an embryo culture-based method for the production and propagation of $F_1$ generation progeny obtained from controlled crosses for OQDS resistance improvement in olive. The initial step involved crossing combining disease-resistant cultivars with other cvs carrying desirable bioagronomic traits.

Fingerprinting based on SSR markers has been used as a tool to support the breeding activities, and it permitted the correct identification of each parental cultivar used as a parental and all the clones analyzed resulted identical to the reference variety.

Subsequently, two different procedures were evaluated to develop hybrid generation. The first procedure relied on the traditional sowing of olive stones into seedbeds. The second relied on the in vitro culture of isolated embryos followed by a multiplication phase through serial subcultures on a Leva MSM [37] modified medium.

Seed germination is an important process in traditional breeding strategies to exploit the genetic variability of a species. However, the germination of olive seeds is prevented due to the inhibitory constraints from the woody endocarp, seed integument, and endosperm leading it to occur at a low rate and over a long time [11,14,15]. Therefore, the removal of woody endocarp, of seed coat, and of endosperm followed by in vitro culture of embryos have been used to speed up the germination in olive [18]. Olive seeds from controlled crosses carried out in 2017 were subjected to direct germination tests in a seedbed and to in vitro culture of embryos, also evaluating the efficiency of two embryo culture mediums. Seeds relative to 2019 crossings were used for embryo culture, exclusively. Sown in seedbeds, the olive stones had a scalar germination that occurred starting from three months, while after five months over 95% of the seedlings had emerged. The germination rate after five months ranged from 14.56% to 24.17% with an average value of 17.10%. The embryo culture allowed the achievement of high levels of germination rate in a range between a minimum of 51.67 to a maximum of 57.22% in the 2017 season and from 52.5 to 54.5% in the 2019 of the olive stones, considering all cross combinations. The time of germination was also significantly reduced with the embryo culture compared to direct sowing. The germination of the isolated embryos started after about five days, on both media tested, with divergence and greening of the cotyledons and radicle elongation considered the first visible signs of germination [16], and the process of embryo growth agreed with the time course of olive embryo in vitro development described in the literature [38]. The highest value of germination was reached after only three weeks, and after eight weeks the plantlets had an adequate development for transplantation and acclimatization.

The results were comparable to those obtained in olive germination tests through embryo culture previously carried out reporting values of germinability of 70–80% [18] or 69% [39] within a few weeks and were mainly limited by the high occurrence of empty olive stones which ranged from 25% to 30% in 2017 and from 28.5% to 31% in 2019. The high rates of seed abortion obtained in the present work could be due to water stress conditions as the seed-carrying plants were grown under a non-irrigated regime [18]. Excluding the aborted seeds, the germinability reached mean values of 75.32% in the 2017 and of 76.15% in the 2019 cultured embryos. Medium 1 allowed the achievement of a high level of germination (91.96%) of the embryos successfully stabilized, similar to that reached

with medium 2 (88.57%) which has already been validated in previous works. Indeed, no significant differences ($p = 0.05$) were found in the germinability of the isolated embryos on the two types of media used, indicating that the new substrate tested represents an effective medium to achieve excellent germination rates of olive embryos with relatively low costs for its hormone-free composition. Moreover, the embryo culture offered the advantage of easily establishing in vitro cultures of the plantlets for further multiplication of the new material in order to carry out evaluations on the relative resistance behavior to *Xf* or for studies aimed to investigate the pathogen–host interaction. The evaluation of this aspect was of particular interest since several olive cultivars have a difficult adaptation to the micropropagation [40,41]. The plantlet response to the clonal multiplication potential trial resulted in shoot multiplication at the average rate of 2.28-fold for each subculture cycle on Leva MSM [37] modified medium. In addition, the explants did not show developmental disorders such as hyperhydricity, yellowing, or leaf abscission. Therefore, this study provides evidence on the possibility of rapidly and easily multiplying, in a limited time and space throughout the year, the plantlets obtained from crossings compared to traditional methods, which would otherwise take longer to propagate the seedlings. The plantlets obtained can also be subjected to a long-term in vitro maintenance, which prevents their infection by pathogens.

The in vitro technique here proposed is an efficient, rapid, and straightforward way to obtain a high olive seed germinability and an adequate number of plants. It is extremely valuable for breeders since it allows them to perform the selection of interest genotypes earlier compared to traditional methods.

## 5. Conclusions

The experience on germination tests of olive seeds from cross-breeding, confirms the technique of embryo culture as a valid tool that allows overcoming drawbacks related to the long dormancy of olive seeds, the non-uniform germination, and the low fruit setting rate, saving the time needed to produce hybrids and contribute to reduce the time for olive genetic improvement programs. Moreover, the plantlets were effectively multiplied with the potential to produce in a short time a large quantity of plant material for their evaluations. The multiplication protocol here proposed is suitable as an integral tool in breeding programs as a means for preserving the selected lines.

**Author Contributions:** V.M., L.S., C.M. and G.B. designed experiments. V.M., L.S., O.P., V.R., A.C., A.S., V.F., P.V. and G.B. performed experiments. L.S., C.M. and G.B. contributed reagents, materials, and analysis tools. V.M. and V.F. analyzed and interpreted the data. V.M., L.S., C.M., V.F. and G.B. wrote and edited the manuscript. All authors have read and agreed to the published version of the manuscript.

**Funding:** This research was funded by Apulian Region-Department of agriculture, rural and environmental development, 70100 Bari, Italy, within the project: "Valutazione del germoplasma olivicolo pugliese e miglioramento genetico per la resistenza a *Xylella fastidiosa*" (Re.d.O.Xy) through D.D. 309 of the 23 September 2016 (Grant No.: B36J16002180007), and promoted by Department of Soil, Plant and Food Sciences (DiSSPA), University of Bari Aldo Moro, 70126 Bari, Italy.

**Institutional Review Board Statement:** Not applicable.

**Informed Consent Statement:** Not applicable.

**Data Availability Statement:** The data presented in this study are available on request from the corresponding author.

**Acknowledgments:** The authors wish to thank Vito Nicola Savino for his efforts in the realization of the project and for his technical and scientific support for the development of the experimental plan.

**Conflicts of Interest:** The authors declare no conflict of interest.

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
