# Peer review of "Embryo Culture, In Vitro Propagation, and Molecular Identification for Advanced Olive Breeding Programs"

_horticulturae, doi:10.3390/horticulturae8010036_

Round 1

Reviewer 1 Report

The authors presented two alternative aspects for olive breeding. Traditional sowing method and in vitro propagation of zygotic embryo were investigated. Controlled crossbreeding activities were carried in 2017 and 2019. Direct sowing was performed only for the seeds obtained from the crosses breeding carried out in 2017 while for in vitro culture two seasons were investigated on two alternative substrates. It is not clear why the authors used two season if they did not compare these seasons. What was the mean of that? Although the results part is written shortly the main remark is condisering on Discussion. This part of the manuscript is very limited and must be significantly improved. A half of a page is certainly not enough for the discussion part. There are numerous literature data describing zygotic embryogenesis. The authors need to explain why they used specifically zygotic embryos for in vitro culture. The results of this work confirmed that in vitro culture techniques reduce the time for olive breeding programs but is it cheaper than traditional and coventional methods?

All of my comments are highlighted in yellow color directly in the text manuscript.

Reviewer 2 Report

The present manuscript presents some good data that would be very useful for olive breeding programs. On the other hand, there are missing important data of in vitro propagation.

(l. 157-158):   …’axillary and adventitious shoots were counted, moreover, the length of the shoots and the number of nodes were determined’. There is also reference about the multiplication rate (l. 159-160). On the other hand I did not find relative tables or figures about in the manuscript.

As regards statistical analysis (l. 175-178), it is missing from the data tables or figures) (i.e Table 2, 3, 4; Figure 2). I think also that l. 240-244 should be written again to present clear the data.

Lines 210-213 present also confused data linked with Table 3.

Hence the data are not clearly presented and the conclusions are not consistent with the evidence and arguments presented.

The manuscript is a good idea and provides some useful data that is not in the current literature, but it should be revised to improve the clarity of the presentation; a clear discussion of the statistical evaluation should be presented as well.

Last but not least is that the title of the manuscript is about embryoculture and in vitro culture of olive seedlings; it does not include DNA analysis and the crossings you did.

Keywords do not include also information about DNA analysis or crossings.

I suggest to the authors to enrich discussion section that does not include discussion about crossings.

In conclusion, the manuscript should be accepted but only after a good revision effort.

Originality

Originality is good; the idea is sound; the data obtained is original.

Methodology: 

The methodology should be enriched by statistical analysis of morphometric data (node number, lenth of shoots etc)

Clarity of Presentation

Τhe data is useful, but it is a bit confusing as presented. Authors should present more clear  the data.

Potential Significance

The potential significance is good.

Overall

Overall, I think this manuscript discusses useful data and has a good objective and hypothesis, but it needs a very good revision effort to be presented as a manuscript in print in HORTICULTURAE.

Round 2

Reviewer 1 Report

The revised version of paper addressed to all reviewer concerns in details and this revised and improved manuscript according to reviewer suggestions is now acceptable for publication.

Reviewer 2 Report

Dear authors,  

I think that the manuscript discusses useful data. Now it is ready to be presented as a printed manuscript in Horticulturae.

This manuscript is a resubmission of an earlier submission. The following is a list of the peer review reports and author responses from that submission.

Round 1

Reviewer 1 Report

The scientific and practical level of the paper is quite high, corresponds to the level of the journal. The text is presented clearly and logically. The level of presentation of the manuscript provides a complete understanding of the rapid protocol of cultivation of isolated embryos in vitro for the production and reproduction of olive seedlings. The disadvantage of the described technique is its complexity, but the possibility of practical implementation using existing technologies eliminates this disadvantage.

For a more complete demonstration of all the advantages of the developed protocol, the Introduction section can be supplemented with information on the practical application of this technique and the results of the study to automate the production of forest reproductive material (in vitro) and positive growth experience in the field. 

I believe that the article can be published in the journal with minor revizions.